# Linking Digital Technologies to Sustainability through Industry 5.0: A bibliometric Analysis

**Adel Ben Youssef** [1,*] and **Issam Mejri** [2]

1    GREDEG-CNRS, Université Côté d'Azur, 5 Rue Du 22ème BCA, 06300 Nice, France
2    IPAG Entrepreneurship & Family Business Center, IPAG Business School, 4 Bd Carabacel, 06000 Nice, France
*    Correspondence: adel.ben-youssef@univ-cotedazur.fr

**Abstract:** Industry 5.0 is a new phase of industrialization which focuses on humans, resilience, and sustainability. The importance of the research on Industry 5.0 has grown considerably and includes a range of different themes. Using a large corpus of data from Scopus, this study conducts a bibliometric review with the aim of providing a holistic overview of the research on Industry 5.0. We review 300 publications on Industry 5.0 to identify their theoretical foundations, research trajectories, and main topics, as well as to propose new research orientations. Our review is based on the integration of a co-citation analysis, historical direct citation analysis, and co-occurrence analysis. We found that most articles on Industry 5.0 have been published during 2020–2023 and focus mainly on India, China, and the United States. IEEE Transactions on Industrial Informatics, Sustainability, and Applied Sciences are the most significant journals publishing research on Industry 5.0. Sustainable development, human-centricity, smart manufacturing, and 6G are the most frequent concepts in the Industry 5.0 domain.

**Keywords:** Industry 5.0; bibliometric analysis; bibliometrix; co-citation analysis; bibliographic coupling





## 1. Introduction

Digital technologies are considered general purpose technologies (GPTs) [1–3] which can affect entire economies and societies worldwide. A GPT is defined as "a technology that initially has many uses and ripple effects as well as opportunities for continuous improvement" [4]. These technologies are not intended to benefit single individuals or fulfill single objectives; rather, they have broad influence and application but require appropriate capabilities to realize their full potential for addressing economic, societal, and environmental issues.

Debate on the impacts of GPTs on our economies, lifestyles, and social ties has been ongoing since [5] proposal of the productivity paradox. The digital transformation increased debate among policymakers, economists, and industry leaders about its societal impact and its effects on jobs, wages, inequality, health, resources efficiency, and security. According to [6], digital transformation refers to the use of a mix of information, computation, communication, and networking technologies to induce significant changes in the characteristics of an entity, thereby improving its overall performance.

There have been successive waves of complementary digital technologies. During the 18th and 19th centuries, Industry 1.0 was focused on manual labor performed by people in industry and agriculture. Between the early 20th century and the 1980s, Industry 2.0 was characterized by advances in mechanical and electrical technologies. The period between 1980 and 2000 saw the emergence of Industry 3.0, characterized by a shift from analog to digital, shorter product life cycles, and modular goods. The advent of Industry 4.0 occurred in 2000, marked by the integration of the Internet of Things (IoT), Big Data, electric vehicles, 3D printing, cloud computing, and artificial intelligence (AI) and is the result of recent

exponential progress in the IoT [7]. Industry 4.0 has similarities with previous industrial revolutions but differs in its effect on industry [8,9].

Following the birth of Industry 4.0 and the rapid digitalization of several economic sectors, Industry 5.0 emerged. Industry 5.0 exploits the potential of the industrial sector to achieve societal goals that go beyond job creation and economic expansion. It promotes sustainable prosperity by ensuring that production respects planetary limits and prioritizes the well-being of industrial workers by placing them at the heart of the manufacturing process [10,11].

New technologies are replacing existing systems and providing incentives for further technological advancements. AI, nanotechnology, quantum computing, synthetic biology, and robots are a few of the technologies which will dwarf the digital advances made since the 1960s and produce unimaginable realities. The business models in all sectors will be disrupted and altered. Manufacturing, logistics, tourism, agriculture, and other industries are among those where Industry 5.0 is predicted to have a major impact. Integration with Industry 4.0 could boost output and efficiency while enabling greater customization and flexibility in service delivery and production processes. Higher levels of automation and data analytics will allow Industry 5.0 to boost productivity and efficiency while enabling greater customization and production flexibility. Industry 5.0 is also anticipated to promote sustainable growth and reduce the negative effects on the environment.

There is a large body of work on the potential of Industry 4.0 and evidence of its impacts, but work on Industry 5.0 is still emerging. There are several surveys of Industry 4.0 [12–16], but we need an overview to understand recent developments in Industry 5.0 (trends, strengths, and gaps). According to the strategy of Industry 5.0 for Europe [17], Industry 4.0 is not the right approach to achieve the 2030 goals. Industry 5.0 truly integrates the Industry 4.0 strategy into a larger context, giving the technological revolution of industrial production a regenerative purpose and directionality for people–planet–prosperity rather than just value extraction to benefit shareholders. Some view the paradigm of Industry 5.0 as an evolutionary approach, incremental advancement that builds on the concepts and practices of Industry 4.0, and others view Industry 5.0 as a complement to the Industry 4.0 paradigm [18].

The present paper reviews the literature on Industry 5.0 based on a sample from Scopus. It uses bibliometric analysis to describe the fields' theoretical foundations, research trajectories, and main topics, as well as to propose new research orientations. Our research question is: "What are the implications, future prospects, and research areas of Industry 5.0?" It contributes to the existing literature by giving information on the Industry 5.0 research, the most authors cited, the most countries that have published on industry, etc. This provides relevant information for the researchers interested in this topic.

The structure of this paper is as follows. In Section 2, the literature on Industry 5.0 is discussed. In Section 3, the methodology of this research is described. In Section 4, the results and key findings are presented. In Section 5, some conclusions and limitations are presented.

## 2. Literature Review: An Overview of Industry 5.0

The term Industry 4.0 was coined in 2011 in Germany at Hanover Fair [19]. Industry 4.0 has been defined in multiple ways. The most widely used is by [20], who characterize it as a new paradigm that uses cyber-physical systems (CPS) technology to establish connections between the digital and physical worlds, facilitating the creation of smart factories and intelligent production processes. These systems facilitate the collection and analysis of data in real time, and greater automation and optimization of manufacturing processes. The integration of AI, the IoT, and other Industry 4.0 technologies enables systems to learn and adapt to changing conditions and make decisions. This allows for greater manufacturing flexibility and customization and the ability to respond quickly to changes in demand.

The term Industry 5.0 was coined by Michael Rada [21]. According to [22], the main distinguishing factor between Industry 4.0 and Industry 5.0 is the increased level

of interaction between humans and machines. This interaction allows individuals to express themselves freely through personalized goods and services. Industry 5.0 allows the provision of more personalized goods and services and is enabled by the participation of users in the design of goods and services. Ozkeser [22] believed that Industry 4.0 is making room progressively for Industry 5.0 over time, due to increased human–machine interaction and a greater emphasis on user- and human-centered goods.

Industry 5.0 is based on three core values: human-centricity, sustainability, and resilience [10]. The first involves the move from technology-driven advancements to a fully human-centered and society-centered approach which puts people's fundamental needs and interests at the center of the production process. Industry workers need to learn new skills and continue their education to improve their career prospects and achieve a good work–life balance. The second is aimed at conserving the planet which will require among other things a circular economy to improve resources' efficiency and effectiveness through reuse, repurposing, and recycling alongside reducing waste and mitigating environmental effects. The third value of resilience is aimed at increasing industrial production robustness to make it resilient to disruption and able to supply and support vital infrastructure in times of emergency.

There is a large body of research that examines and contrasts the advantages and disadvantages of Industry 5.0, the enabling of technology, and industrial applications. Akundi [18] listed five main themes framing Industry 5.0 research, namely (i) supply chain assessment and improvement, (ii) business advancement and digitization, (iii) implementation of smart and green manufacturing processes, (iv) changes driven by the Internet of Things (IoT), artificial intelligence (AI), and Big Data, and (v) connectivity between humans and machines. These themes indicate potential future research topics and suggest the greater interest of the scientific community in Industry 5.0 as a bridge to human–machine interaction.

Aslam et al. [23] provide a comprehensive literature review and propose an "Absolute Innovation Management" framework. To prepare firms for the IoT and Industry 5.0 transformation, integrated innovation, design thinking, and strategy will be required. Ben Youssef and Zeqiri [9] summarizes the characteristics of Industry 5.0 in a literature review and a discussion of enabling technologies, suggesting that Industry 5.0 is aimed at optimizing human and machine productivity and discusses the related problems. Ruiz-de-la-Torre et al. [24] present a bibliometric study of the Industry 5.0 scientific–technological field, identifying the beginning of its growth stage, visualizing the leading countries and the most cited authors and contributions, and obtaining the general overview of the field of study.

Ivanov [25] identifies elements that define Industry 5.0 as an organizational and technology framework. The most significant technology drivers of Industry 5.0 are identification, automation, data analytics processing, collaboration, coordination, and communication within a focus on the areas of management, technology, performance evaluation, and organization. He highlights also that Industry 5.0 involves the societal, network, and plant levels, and he proposes the concept of a triple bottom line as comprising resilient value creation, human well-being, and a sustainable society.

Xu et al. [11] address the conceptualization, perception, and inception of Industry 4.0 and Industry 5.0. They suggest that while Industry 4.0 is tech-driven, Industry 5.0 is value-driven and involves fundamental societal needs, values, and responsibilities and depends on the output of Industry 4.0 for technological advancements and technical fixes. Huang et al.'s [26] paper distinguishes between Society 5.0 and Industry 5.0 and highlights the need to understand both in order to achieve advancements in related theories, approaches, and applications. According to [26], Industry 5.0 and Society 5.0 differ in terms of goals, values, organization, and technology.

An in-depth review of the effects of new manufacturing industries and skills is provided by [27] who suggest that Industry 5.0 enhances cooperation between humans and robots and advocates for use of humans and robots together in the manufacturing process.

Nahavandi [28] outlines how thinking about robots becomes automatic and states that robots then function as collaborators rather than rivals. He claims that the effects of Industry 5.0 on the manufacturing sector and the economy more generally should be analyzed from an economic and productivity point of view, and he argues that Industry 5.0 will generate more employment than it eliminates.

Kolade and Owseni's [29] view highlights three major theoretical research stances: socio-technical systems theory, skill-biased technological change which includes variations such as task-biased and routine-biased technological change, and the political economy of digital transformation. They suggest that the most promising path for future research would be to merge these three perspectives into a unified and coherent theoretical framework on the future complicated problems and multifaceted opportunities brought by Employment 5.0.

To boost the effectiveness of digital green innovation and business competitiveness, [30] investigated how digital technology could be used in industrial processes. They highlight the intricate nature of the digital green innovation process and propose a theoretical framework to allow businesses to choose knowledge partners that are compatible with their specific environment and workflow. Their study should allow businesses to become more competitive and to survive and grow in the current business climate.

## 3. Research Methodology

*Data Collection*

We chose to collect our data from the Scopus database. We opted for Scopus, first, because it is among the largest peer-reviewed databases of multidisciplinary research publications; second, because it is easily accessible and includes the most reputable journals in the fields of economics, business, and engineering; third, it allows for advanced search and export of bibliographical data based on specific objectives; and fourth, compared to the Web of Science (WOS-SSCI), for instance, it includes a significant number of articles relevant to our study. Figure 1 presents the methodology used.

To collect the data, we used the following search string within the document topic field:

"Industry 5.0" OR "artificial intelligence" OR "smart manufacturing" OR "big data" OR "internet of things" OR "human-machine coexistence" OR "Smart Sustainability" AND "Industry 5.0".

The research objectives depicted in Table 1 are based on the analytical method used, type of data, and technology employed in the research, as well as the analytical strategy, analytical tool, data type, and technology. First, we conduct a performance analysis of Industry 5.0 research followed by an analysis of the most dominant methodologies and geographical focuses in Industry 5.0 research. We then provide a science mapping of Industry 5.0 research.

Table 2 presents some baseline statistics related to the methodology used. The study spans the years 2016 to 2022 and includes 170 literature sources (journals, books, etc.) which in turn include 300 documents—233 articles, 5 books, 19 book chapters, 5 editorials, and 38 reviews.

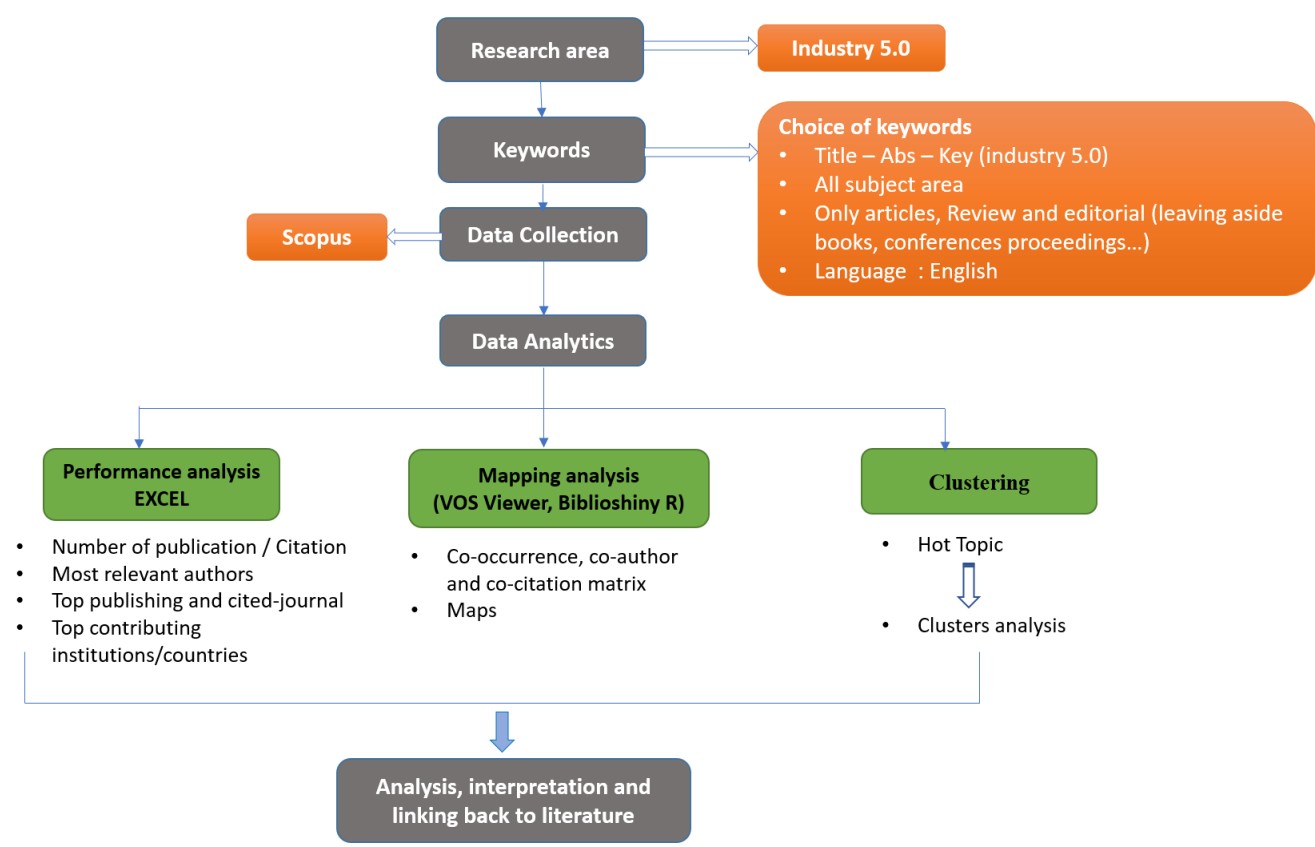

**Figure 1.** Methodological protocol applied in Scopus for extracting articles.

**Table 1.** Mapping of research objectives to methodology.

| Research Objectives | Analytical Method | Type of Data | Technologies Employed |
|---|---|---|---|
| Present a performance analysis of Industry 5.0 research. | Performance analysis | Publications<br>● Citations<br>● h-index | ● Scopus |
| Provide an analysis of the most dominant methodologies and geographical focuses in Industry 5.0 research. | Descriptive analysis | Full text (methodology, sample country) | ● VOSviewer<br>● R Studio (bibliometrix) |
| Provide a science mapping of Industry 5.0 research. | ● Co-authorship analysis<br>● Co-citation analysis<br>● Bibliographic coupling | ● Authors<br>● References | |

**Table 2.** Baseline statistics (source: bibliometrix using data from Scopus).

| Description | Results |
|---|---|
| Main Information About Data | |
| Timespan | 2016–2022 |
| Sources (Journals, Books, etc.) | 170 |
| Documents | 300 |
| Annual Growth Rate % | 144.95 |
| Document Average Age | 1.42 |

**Table 2.** *Cont.*

| Description | Results |
| --- | --- |
| Average Citations per Doc | 8.62 |
| References | 20,714 |
| Document Content | |
| Keywords Plus (Id) | 1770 |
| Author Keywords (De) | 1192 |
| Authors | |
| Authors | 1061 |
| Authors of Single-Authored Docs | 23 |
| Co-authorships | |
| Single-Authored Docs | 23 |
| Co-Authors per Doc | 4.12 |
| International Co-Authorships % | 42.33 |
| Document Types | |
| Article | 233 |
| Book | 5 |
| Book Chapter | 19 |
| Editorial | 5 |
| Review | 38 |

## 4. Results

A bibliometric analysis was conducted on our sample using two main methods: descriptive and performance analysis, and a mapping technique [31].

### 4.1. Descriptive and Performance Analysis

4.1.1. Scientific Production

Scientific production on Industry 5.0 has grown significantly and especially since 2020. This rapid increase in the number of articles is due to the fact that several new journals have appeared which publish academic work on digitalization, Industry 4.0, and Industry 5.0. In addition, many journals have become more willing to publish work on these areas due to the increasing importance of Industry 5.0 and the digital transformation dynamics around the world.

Figure 2 shows the evolution of scientific production in the field of Industry 5.0 from 2016 to 2022. We observe two phases of development of Industry 5.0 research: a first phase during 2016 to 2020, and an expansion phase in 2020–2022. The year 2020 was pivotal in the development of research on Industry 5.0 when it became a recognized concept along with its enablers and components. In terms of citations, 2022 saw the highest number with 1873 total citations (see Table 3).

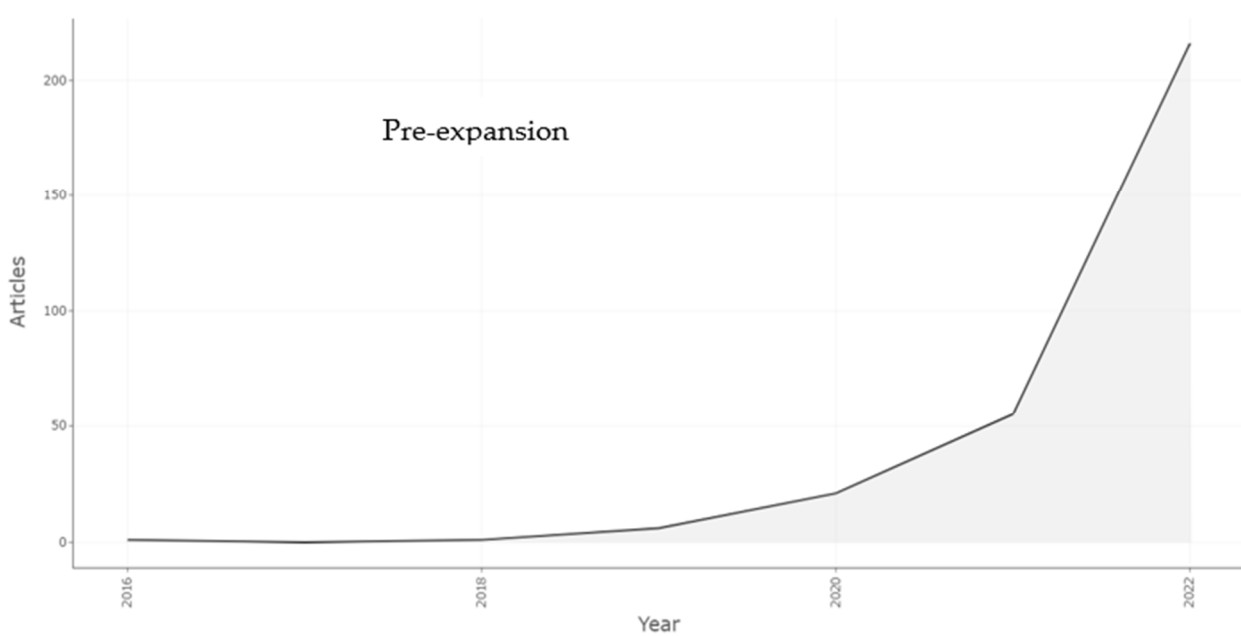

**Figure 2.** Distribution of scientific publications over time. Source: authors' own elaboration.

**Table 3.** Annual citations related to Industry 5.0.

| Year | N | TC |
|---|---|---|
| 2016 | 1 | 0 |
| 2017 | 0 | 0 |
| 2018 | 1 | 9 |
| 2019 | 6 | 34 |
| 2020 | 21 | 124 |
| 2021 | 55 | 432 |
| 2022 | 216 | 1873 |

Source: Authors.

4.1.2. Most Relevant Countries, Journals, Authors, Articles, and Institutions

Bibliometric citation analysis provides an overview of the most relevant elements of countries, journals, articles, authors, and institutions.

India accounts for the majority of articles (53) followed by China (49), and the USA (36). After that, scientific production is shown to be dispersed across different world regions and including developed country authors (Australia, UK, Italy, Spain, Sweden, Canada, etc.) and authors from emerging countries (Saudi Arabia, Pakistan, Indonesia, Poland, etc.) (see Figure 3). Publication on Industry 5.0 by India, China, and the USA received the highest number of citations (Table 4).

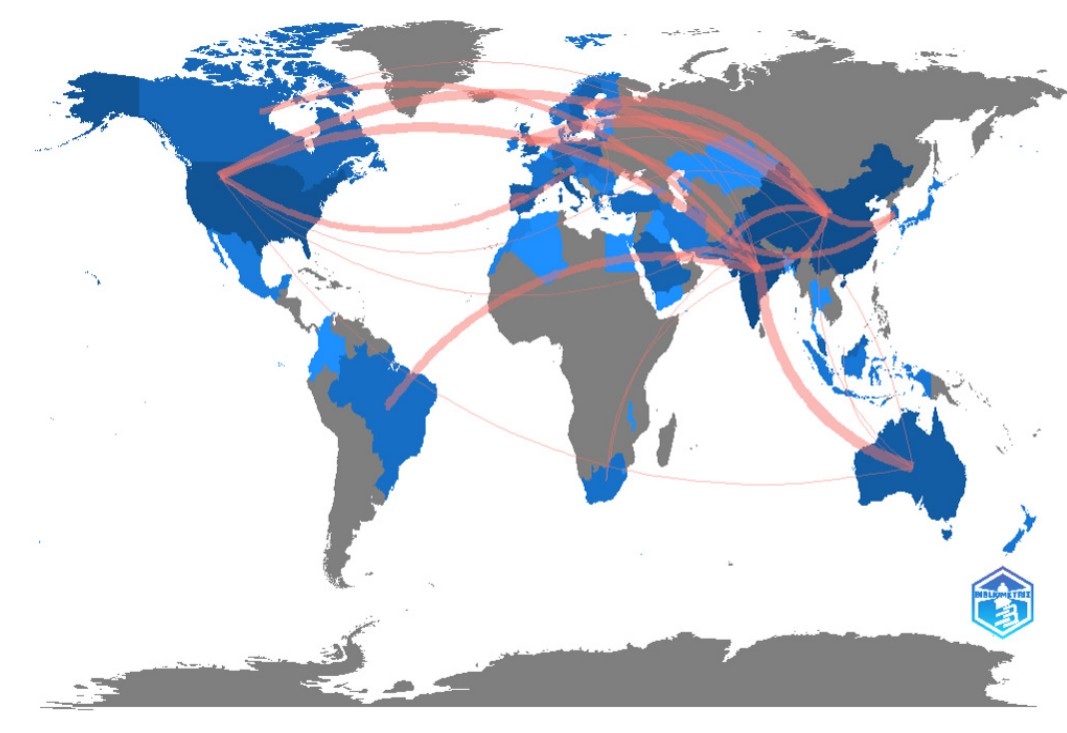

**Figure 3.** Country scientific production.

**Table 4.** Most cited countries for work on Industry 5.0.

| Country | Documents | Total Citations | Total Link Strength |
|---|---|---|---|
| India | 53 | 676 | 179 |
| China | 49 | 656 | 252 |
| United States | 36 | 492 | 165 |
| Italy | 28 | 208 | 119 |
| United Kingdom | 24 | 301 | 93 |
| Australia | 21 | 366 | 173 |
| Spain | 19 | 94 | 68 |
| Portugal | 17 | 47 | 58 |
| Pakistan | 15 | 139 | 22 |
| Poland | 14 | 54 | 77 |
| Sweden | 14 | 299 | 195 |
| Malaysia | 13 | 95 | 91 |
| Saudi Arabia | 13 | 75 | 49 |
| Canada | 12 | 230 | 25 |
| South Korea | 12 | 293 | 59 |
| Germany | 11 | 165 | 88 |
| Greece | 11 | 56 | 93 |
| Brazil | 8 | 15 | 57 |
| Turkey | 8 | 219 | 24 |

**Table 4.** *Cont.*

| Country | Documents | Total Citations | Total Link Strength |
|---|---|---|---|
| Lithuania | 7 | 54 | 81 |
| Slovakia | 7 | 22 | 38 |
| United Arab Emirates | 7 | 17 | 1 |
| France | 6 | 15 | 13 |
| Norway | 6 | 35 | 17 |
| Russian Federation | 6 | 32 | 5 |
| Slovenia | 6 | 24 | 23 |
| South Africa | 6 | 198 | 39 |
| Taiwan | 6 | 80 | 16 |
| Hungary | 5 | 5 | 29 |
| Romania | 5 | 11 | 9 |

Abbreviations: TC, Total citation VOSviewer (10 January 2023); TP, Total publications; TLS, Total link strength.

### 4.1.3. Most Productive and Influential Journals

Tables 5 and 6 present the most important journals publishing on Industry 5.0. These are derived from a ranking based on the number of papers published on Industry 5.0 and total citations. IEEE Transactions on Industrial Informatics, Sustainability (Switzerland), and Applied Sciences (Switzerland) are the three journals receiving the highest number of citations to Industry 5.0 publications.

**Table 5.** Top ten most productive journals publishing on Industry 5.0.

| R | Journals | NP | TC | h_index | g_index | m_index | PY_start |
|---|---|---|---|---|---|---|---|
| 1 | IEEE Transactions on Industrial Informatics | 23 | 113 | 6 | 10 | 2 | 2021 |
| 2 | Sustainability (Switzerland) | 16 | 329 | 6 | 16 | 1.2 | 2019 |
| 3 | Applied Sciences (Switzerland) | 12 | 122 | 4 | 11 | 1 | 2020 |
| 4 | Sensors | 11 | 49 | 4 | 7 | 1.333 | 2021 |
| 5 | Journal of Manufacturing Systems | 8 | 206 | 5 | 8 | 1.667 | 2021 |
| 6 | Energies | 8 | 36 | 3 | 6 | 1 | 2021 |
| 7 | Journal of The Knowledge Economy | 7 | 100 | 6 | 7 | 2 | 2021 |
| 8 | Computers and Industrial Engineering | 7 | 35 | 4 | 5 | 1.333 | 2021 |
| 9 | IEEE Access | 6 | 16 | 2 | 4 | 0.667 | 2021 |
| 10 | International Journal of Production Research | 5 | 16 | 3 | 4 | 1.5 | 2022 |

Sustainability (Switzerland) is ranked first with 329 citations to 16 articles published between 2019 and 2022 followed by Journal of Industrial Information Integration with 206 citations and Journal of Manufacturing Systems with 206 citations (Table 6). These citation counts reflect the number of citations to the articles included in our study identified via our keyword searches.

**Table 6.** Top ten most cited journals publishing on Industry 5.0.

| R | Journals | NP | TC | h_index | g_index | m_index | PY_start |
|---|---|---|---|---|---|---|---|
| 1 | Sustainability (Switzerland) | 16 | 329 | 6 | 16 | 1.2 | 2019 |
| 2 | Journal of Industrial Information Integration | 4 | 208 | 3 | 4 | 1.5 | 2022 |
| 3 | Journal of Manufacturing Systems | 8 | 206 | 5 | 8 | 1.667 | 2021 |
| 4 | OMICS A Journal of Integrative Biology | 2 | 183 | 2 | 2 | 0.333 | 2018 |
| 5 | Applied Sciences (Switzerland) | 12 | 122 | 4 | 11 | 1 | 2020 |
| 6 | Journal of Industrial Integration and Management | 3 | 117 | 2 | 3 | 0.5 | 2020 |
| 7 | IEEE Transactions on Industrial Informatics | 23 | 113 | 6 | 10 | 2 | 2021 |
| 8 | Journal of The Knowledge Economy | 7 | 100 | 6 | 7 | 2 | 2021 |
| 9 | IEEE Internet of Things Journal | 2 | 92 | 2 | 2 | 0.5 | 2020 |
| 10 | Information Systems Frontiers | 2 | 87 | 2 | 2 | 0.5 | 2020 |

### 4.1.4. Most Productive and Influential Authors

Tables 7 and 8 and Figure 4 show that the most productive and influential authors on industry 5.0 are Carayannis from George Washington University with seven papers in 2021, followed by Campbell, Chang, Dev, Grabowska, Li, Lu, Saniuk, and Zhang all with three papers, and Haleem with two papers. The most cited author writing about Industry 5.0 is Nahavandi from Deakin University with 146 citations, followed by Hekim with 139 citations.

**Table 7.** Most productive top authors.

| R | Authors | Current Affiliation * | Country | TP | TC | TLS | H | G | M | PY_start |
|---|---|---|---|---|---|---|---|---|---|---|
| 1 | Carayannis, Elias G. | George Washington University | USA | 7 | 52 | 60 | 5 | 7 | 2.5 | 2021 |
| 2 | Campbell, David F.J. | Danube University Krems | Austria | 3 | 27 | 38 | 2 | 3 | 1 | 2021 |
| 3 | Chang, Victor | Teesside University | United Kingdom | 3 | 73 | 2 | 2 | 3 | 0.667 | 2020 |
| 4 | Dev, Kapal | Munster Technological University | Ireland | 3 | 52 | 27 | 2 | 2 | 2 | 2022 |
| 5 | Grabowska, Sandra | Silesian University of Technology | Poland | 3 | 2 | 43 | 1 | 1 | 1 | 2022 |
| 6 | Li, Xingwang | Henan Polytechnic University | China | 3 | 3 | 12 | 1 | 1 | 1 | 2022 |
| 7 | Lu, Yuqian | The University of Auckland | New Zealand | 3 | 26 | 50 | 1 | 3 | 0.5 | 2021 |

**Table 7.** *Cont*.

| R | Authors | Current Affiliation * | Country | TP | TC | TLS | H | G | M | PY_start |
|---|---------|----------------------|---------|----|----|-----|---|---|---|----------|
| 8 | Saniuk, Sebastian | University of Zielona Gora | Poland | 3 | 2 | 43 | 1 | 1 | 1 | 2022 |
| 9 | Zhang, Qinzi | Boston College | USA | 3 | 1 | 2 | 1 | 1 | 1 | 2022 |
| 10 | Haleem, Abid | Jamia Millia Islamia | India | 2 | 71 | 78 | 2 | 2 | 0.667 | 2020 |

\* Information gathered on Google Scholar and ResearchGate platforms.

**Table 8.** Most cited authors.

| R | Authors | Current Affiliation * | Country | TP | TC | TLS | H | G | M | PY_Start |
|---|---------|----------------------|---------|----|----|-----|---|---|---|----------|
| 1 | Nahavandi, Saeid | Deakin University | Australia | 1 | 146 | 139 | 1 | 1 | 0.25 | 2019 |
| 2 | Hekim, Nezih | Biruni Üniversitesi | Turkey | 2 | 139 | 22 | 1 | 2 | 0.2 | 2018 |
| 3 | Özdemir, Vural | Gaziantep University | Turkey | 2 | 139 | 22 | 1 | 2 | 0.2 | 2018 |
| 4 | Chang, Victor | Teesside University | United Kingdom | 3 | 73 | 2 | 2 | 3 | 0.67 | 2020 |
| 5 | Haleem, Abid | Jamia Millia Islamia | India | 2 | 71 | 78 | 2 | 2 | 0.67 | 2020 |
| 6 | Javaid, Mohd | Jamia Millia Islamia | India | 2 | 71 | 78 | 2 | 2 | 0.67 | 2020 |
| 7 | Abdel-Basset, Mohamed | Zagazig University | Egypt | 1 | 68 | 2 | 1 | 1 | 0.33 | 2020 |
| 8 | Gamal, Abduallah | Zagazig University | Egypt | 1 | 68 | 2 | 1 | 1 | 0.33 | 2020 |
| 9 | Manogaran, Gunasekaran | Universidad Distrital Francisco José de Caldas | Colombia | 1 | 68 | 2 | 1 | 1 | 0.33 | 2020 |
| 10 | Carayannis Elias G. | George Washington University | USA | 7 | 52 | 60 | 5 | 7 | 2.5 | 2021 |

\* Information gathered on Google Scholar and ResearchGate platforms.

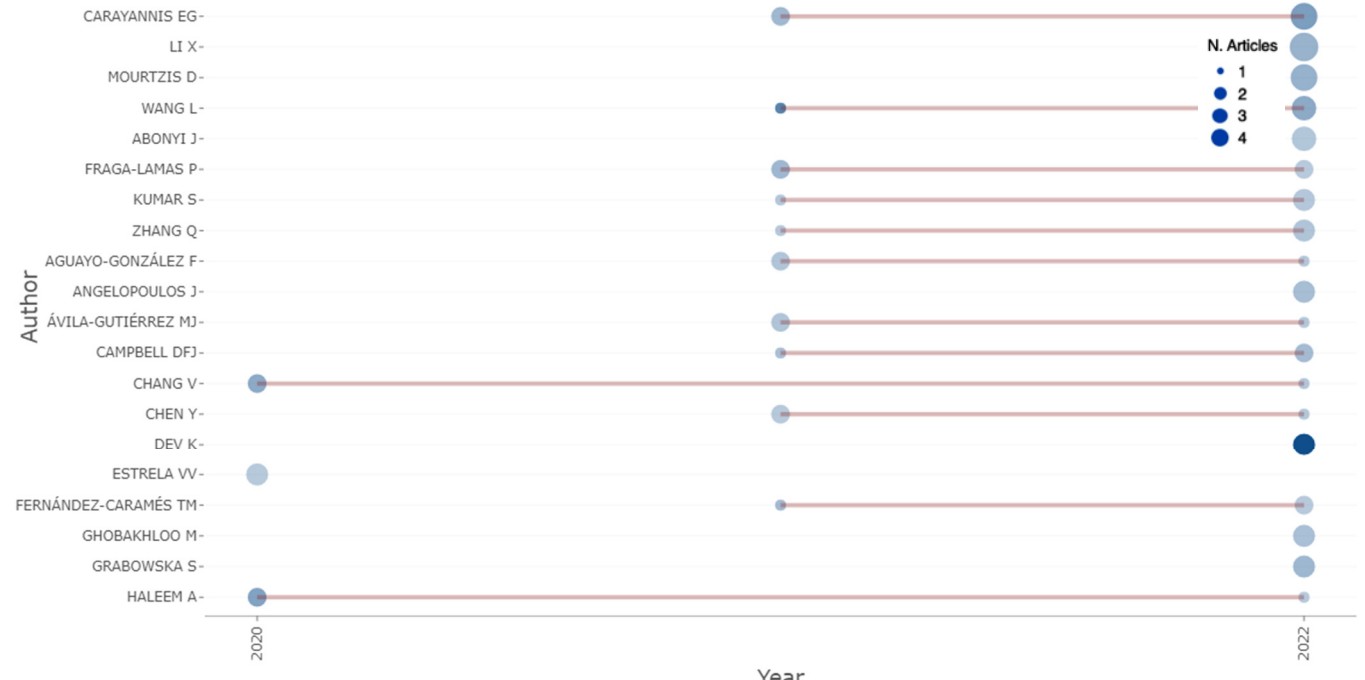

**Figure 4.** Author productivity (output) over time.

### 4.1.5. Most Influential Articles

Table 9 presents the top 20 most cited articles published since 2010. Using year of publication rather than number of citations reduces the bias against more recent articles and is common practice in the literature [32].

**Table 9.** Worldwide 20 most-cited articles on Industry 5.0 (source: VOSviewer using data from Scopus).

| | Paper | DOI | Year | Source | TC | TC Per YEAR | Normalized TC |
|---|---|---|---|---|---|---|---|
| 1 | Nahavandi S. | Industry 5.0—a human-centric solution | 2019 | Sustainability (Switzerland) | 261 | 52.2 | 5.17 |
| 2 | Özdemir V., Hekim N. | Birth of Industry 5.0: Making Sense of Big Data with Artificial Intelligence, "the Internet of Things" and Next-Generation Technology Policy | 2018 | OMICS A Journal of Integrative Biology | 180 | 30 | 1 |
| 3 | Maddikunta P.K.R., Pham Q.-V., B P., Deepa N., Dev K., Gadekallu T.R., Ruby R., Liyanage M. | Industry 5.0: A survey on enabling technologies and potential applications | 2022 | Journal of Industrial Information Integration | 173 | 86.5 | 41.85 |
| 4 | Xu X., Lu Y., Vogel-Heuser B., Wang L. | Industry 4.0 and Industry 5.0—Inception, conception and perception | 2021 | Journal of Manufacturing Systems | 152 | 50.67 | 13.46 |
| 5 | Longo F., Padovano A., Umbrello S. | Value-oriented and ethical technology engineering in industry 5.0: A human-centric perspective for the design of the factory of the future | 2020 | Applied Sciences (Switzerland) | 93 | 23.25 | 3.62 |
| 6 | Abdel-Basset M., Manogaran G., Gamal A., Chang V. | A Novel Intelligent Medical Decision Support Model Based on Soft Computing and IoT | 2020 | IEEE Internet of Things Journal | 89 | 22.25 | 3.46 |
| 7 | Bednar P.M., Welch C. | Socio-Technical Perspectives on Smart Working: Creating Meaningful and Sustainable Systems | 2020 | Information Systems Frontiers | 81 | 20.25 | 3.15 |
| 8 | Pillai S.G., Haldorai K., Seo W.S., Kim W.G. | COVID-19 and hospitality 5.0: Redefining hospitality operations | 2021 | International Journal of Hospitality Management | 78 | 26 | 6.91 |
| 9 | Javaid M., Haleem A., Singh R.P., Ul Haq M.I., Raina A., Suman R. | Industry 5.0: Potential applications in COVID-19 | 2020 | Journal of Industrial Integration and Management | 62 | 15.5 | 2.41 |
| 10 | Javaid M., Haleem A. | Critical components of Industry 5.0 towards a successful adoption in the field of manufacturing | 2020 | Journal of Industrial Integration and Management | 54 | 13.5 | 2.1 |

**Table 9.** *Cont.*

| | Paper | DOI | Year | Source | TC | TC Per YEAR | Normalized TC |
|---|---|---|---|---|---|---|---|
| 11 | Aslam F., Aimin W., Li M., Rehman K.U. | Innovation in the era of IoT and industry 5.0: Absolute innovation management (AIM) framework | 2020 | Information (Switzerland) | 53 | 13.25 | 2.06 |
| 12 | Sachsenmeier P. | Industry 5.0—The Relevance and Implications of Bionics and Synthetic Biology | 2016 | Engineering | 49 | 6.13 | 1 |
| 13 | ElFar O.A., Chang C.-K., Leong H.Y., Peter A.P., Chew K.W., Show P.L. | Prospects of Industry 5.0 in algae: Customization of production and new advance technology for clean bioenergy generation | 2021 | Energy Conversion and Management: X | 43 | 14.33 | 3.81 |
| 14 | Wang Y., Cai Z., Zhan Z.-H., Zhao B., Tong X., Qi L. | Walrasian Equilibrium-Based Multiobjective Optimization for Task Allocation in Mobile Crowdsourcing | 2020 | IEEE Transactions on Computational Social Systems | 40 | 10 | 1.56 |
| 15 | Jan M.A., Khan F., Khan R., Mastorakis S., Menon V.G., Alazab M., Watters P. | Lightweight Mutual Authentication and Privacy-Preservation Scheme for Intelligent Wearable Devices in Industrial-CPS | 2021 | IEEE Transactions on Industrial Informatics | 35 | 11.67 | 3.1 |
| 16 | Lu Y., Zheng H., Chand S., Xia W., Liu Z, Xu X., Wang L., Qin Z. and Bao J. | Outlook on human-centric manufacturing towards Industry 5.0 | 2022 | Journal of Manufacturing Systems | 30 | 15 | 7.26 |
| 17 | Bhat Sa | Agriculture-Food Supply Chain Management Based on Blockchain and IoT: A Narrative on Enterprise Blockchain Interoperability | 2022 | AGRIC | 30 | 15 | 7.26 |
| 18 | Fraga-Lamas P., Lopes S.I. and Fernández-Caramés T.M. | Green IoT and Edge AI as Key Technological Enablers for a Sustainable Digital Transition towards a Smart Circular Economy: An Industry 5.0 Use Case | 2021 | SENSORS | 30 | 10 | 2.66 |
| 19 | Javed Ar, Shahzad F., Rahmen Su, Zikria Y.B., Razzak I, Jalil Z. and Xu G | Future smart cities: requirements, emerging technologies, applications, challenges, and future aspects | 2022 | CITIES | 28 | 14 | 6.77 |
| 20 | Gürdür Broo D., Kaynak O. and Sait S.M. | Rethinking engineering education at the age of industry 5.0 | 2022 | Journal of Industrial Information Integration | 28 | 14 | 6.77 |

Source: Authors' own elaboration.

### 4.2. Network Analysis

Visualization is a recognized tool for examining complex and massive bibliographic networks in terms of paper co-citations, author keyword co-occurrence, and bibliographic

coupling networks [33]. Visualization is enabled by software tools such as VOSviewer which provides a user-friendly interface.

We examine the researcher bibliographic coupling network, the journal co-citation network, and the author keyword co-occurrence network which will allow us to group our findings and identify sub-themes for future thematic research. The results of the visualization provide a clear picture of the most influential writers and publications in the field.

The network analysis uses VOSviewer software [33] augmented by R studio [34] to reveal the main Industry 5.0 clusters based on conceptual structure.

### 4.2.1. Co-Citation Analysis

The map in Figure 5 indicates the concomitant citations among the journals. Regarding the co-citation of the journals cited on Industry 4.0, Sustainability shows the most important network, followed by IEEE access, Procedia Cirp, etc.

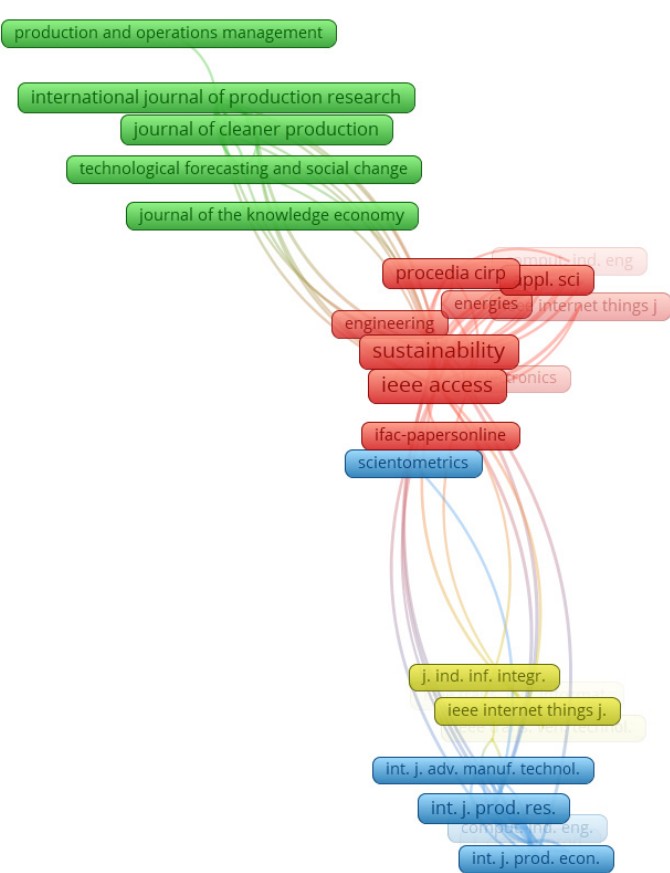

**Figure 5.** Co-citation of journal cited in industry 5.0 field (with VOS Viewer).

### 4.2.2. Bibliographic Coupling

Figure 6 depicts the network of bibliographic couplings in the Industry 5.0 field. The nodes represent documents, and the edges represent bibliographic couplings.

Bibliographic coupling uses the VOSviewer which enables powerful exploration of big datasets and offers a variety of creative data visualization features [33]. China, India, and the US seem to be the most important networks of bibliographic couplings in the Industry 5.0 field.

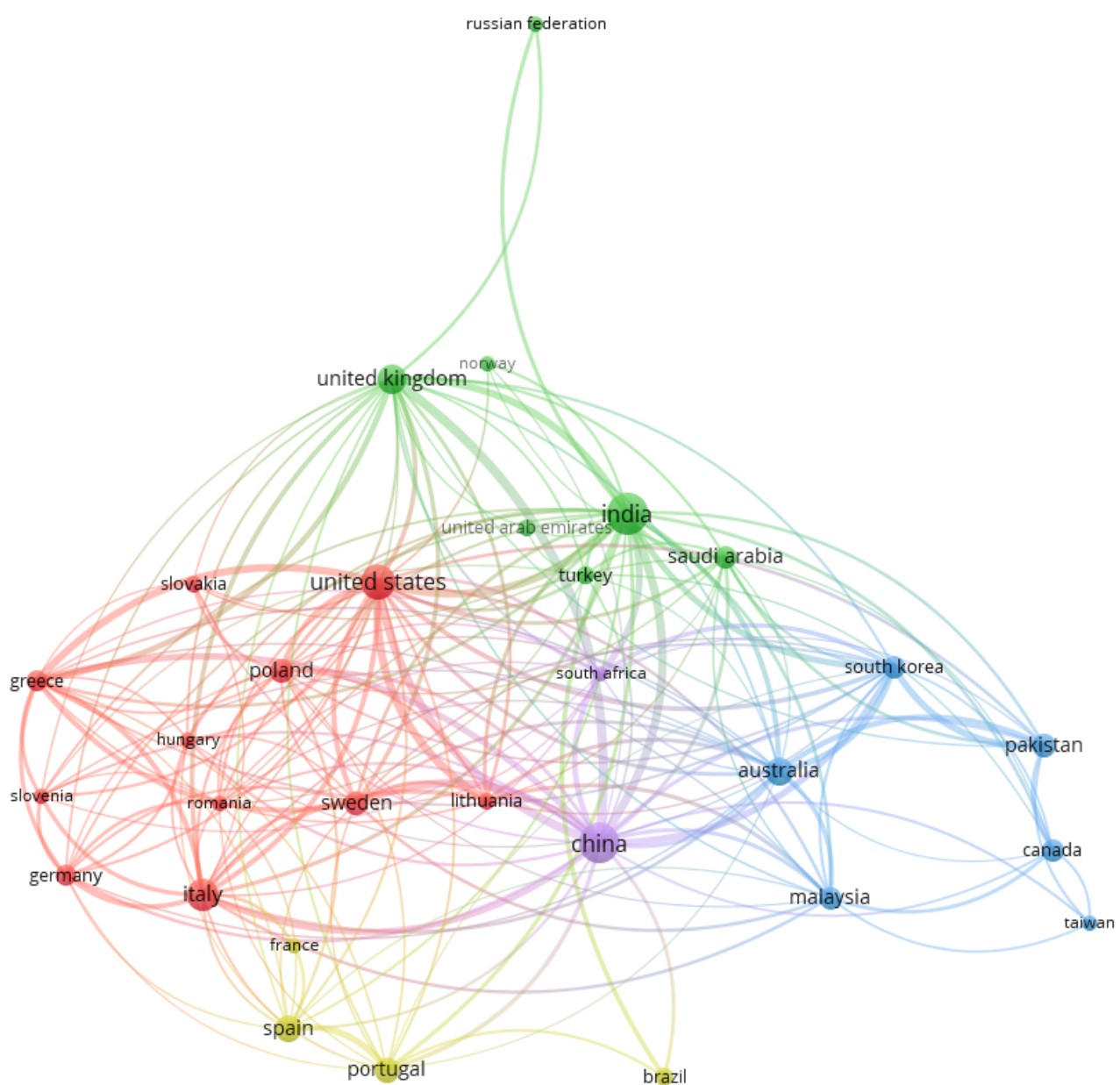

**Figure 6.** Bibliographic coupling of countries publishing in Industry 5.0 field.

*4.3. Co-Word (Or Keyword Co-Occurrence) Analysis*

Cluster Definition and Analysis

In order to determine the main Industry 5.0 thematic clusters, we used the abstracts of the sample publications to construct a co-occurrence network. Figure 7 shows the co-occurrence network of terms. Figure 8 depicts how the terms evolve through the network. The two figures were generated by using the VOSviewer. This co-occurrence network includes four groups that are the most mature elements in Industry 5.0: sustainable development, human-centricity, smart manufacturing, and 6G.

Within the following, we introduce these four themes that were revealed through the analysis of co-occurrence networks and suggest some research directions for each theme.

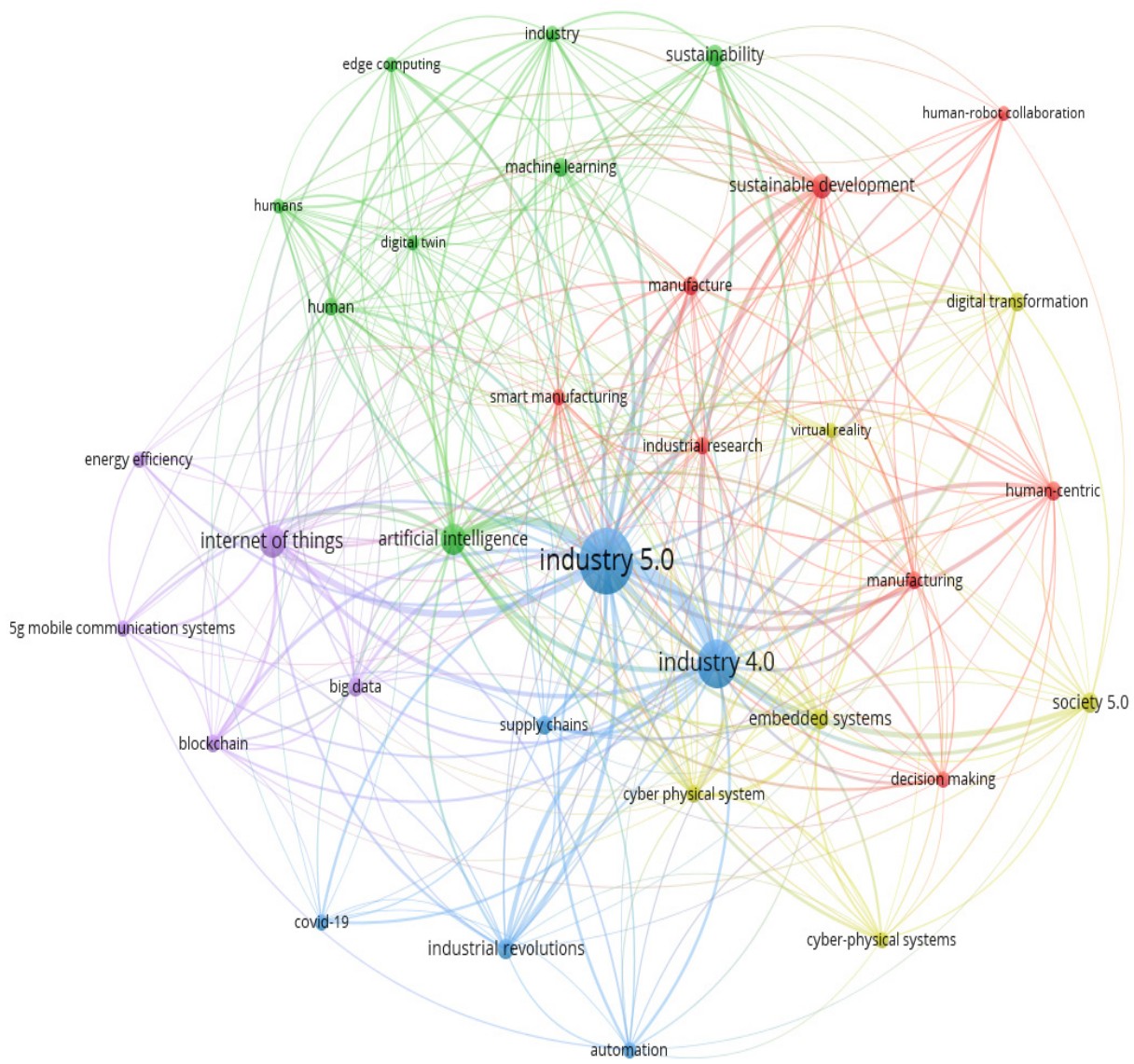

**Figure 7.** All keyword co-occurrences.

### 4.4. Conceptual Structure and Cluster Overview

Cobo et al. [35] present a method for evaluating and visualizing topics in a given research branch. They use thematic maps that provide a clear graphical representation of the topics evaluated in a quadrant. Centrality (x axis) and density (y axis) frame the topic maps. Density measures the development of the selected topic, while centrality measures the importance of the central topic. The thematic map is divided into four parts. Themes located in the lower left part of the map are declining or emerging themes, which may be abandoned or developed by researchers. Themes in the lower right portion of the thematic map are basic patterns that have been extensively researched. Themes in the upper left portion of the map are niche themes that are developed in isolation, while themes in the upper right portion of the map are widely developed. The map shows that there is an important body of research on building information modeling (bim) and digital twins. Decision-making regarding the adoption of Industry 5.0. is an emerging theme while Artificial Intelligence, Internet of Things, etc. are basic themes of Industry 5.0. Digital economy and digital innovation are niche areas which need to be developed (Figure 9).

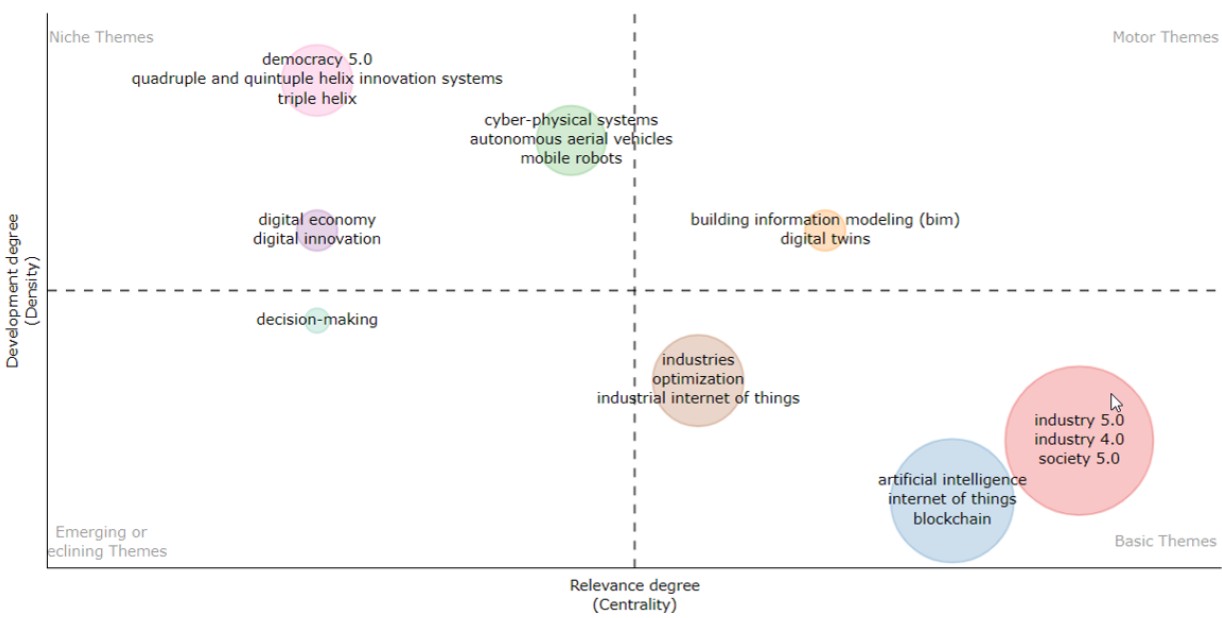

**Figure 8.** Keyword overlay visualization.

**Figure 9.** Thematic map. Source: authors using Biblioshiny.

## 5. Conclusions

Using bibliometric analysis, this study has provided an in-depth elucidation of the geographic scope, methods choices, major topics, and future research objectives in this area using a sizable corpus taken from Scopus. To pinpoint theoretical foundations, research paths, thematic trends, and potential areas for future research, we reviewed 300 papers on Industry 5.0 and conducted co-citations analysis, historical direct citations analysis, and co-occurrence analysis. To analyze our quantitative results, we conducted a qualitative review.

### 5.1. Concluding Comments

Our results point to five main findings.

First, the majority of articles on Industry 5.0 were published since 2016. In terms of advancing Industry 5.0 as a concept and identifying its enablers and components, we took year 2020 as a turning point in the development of research on this topic. Since then, Industry 5.0 has attracted more research attention. This reflects greater concern about society and the environment and the focus on resilience induced by the COVID-19 pandemic. Several fora such as the World Economic Forum called for a reset following the COVID-19 pandemic and take the view that the world needs more resilience-centered, human-centered, and sustainability-driven technologies (Industry 5.0).

Second, most articles (53) were published in India, followed by China (49), and the United States (36). All three countries need a shift in their economies to a more sustainable pathway. All need a new development paradigm centered more on sustainability and technology use to protect employment. The number of their publications also reflects the vitality of their scientific systems to address more advanced thematic topics such as Industry 5.0. However, while papers published in China and India tend to be focused on theoretical perspectives, United States papers tend to be empirical.

Third, IEEE Transactions on Industrial Informatics, Sustainability, and Applied Sciences are the most frequent publishers and most cited journals in the area of Industry 5.0.

Fourth, sustainable development, human-centricity, smart manufacturing, and 6G are well-established concepts in the Industry 5.0 domain, while eco-innovation, communication, SMEs, environment, and mechanisms are more recent topics according to the development of terms within the co-occurrence network.

Fifth, the literature review shows the links with Industry 4.0 and its related technologies (AI, Blockchain, Industrial IoT, etc.), and sustainability and sustainable development. However, the issues of labor consequences including wages, work conditions, job satisfaction, and employment receive less attention. There is a need to address this gap since the move from Industry 4.0. is a shift to more human-centric technologies, making labor substitution and labor complementarity with the technology's urgent issues.

### 5.2. Limitations and Future Research

The current study offers a broad spectrum of research on Industry 5.0. However, more literature is required to homogeneously define Industry 5.0.

Our work has two limitations. First, we use data from only one database; other sources might offer alternative views of the development of Industry 5.0. Second, our research has mainly explored only the dimension of Industry 5.0 as a broad terminology, while several international institutions are tackling this issue under alternative terminologies such as "twin transitions" or "smart and sustainable industry".

Industry 5.0 represents a significant shift in how we should think about industrial processes and the roles of technology and human workers in these industries. A combination of technology and human resources will give Industry 5.0 the potential to create more sustainable, efficient, and human-centric industries to address the 21st century's problems.

We present some orientations for future research. First, future research could focus on better exploring the research on how humans and machines are collaborating in different sectors. Second, research could focus on how Industry 5.0 could help with more sustainable

production processes, such as using renewable energy sources and reducing waste. Third, in the future, researchers should focus on the new cybersecurity solutions to protect Industry 5.0 systems against cyber threats. Fourth, it would be interesting to explore ethical considerations of Industry 5.0, such as the impact on employment and social justice. Fifth, research could focus on developing more advanced twin transition and Industry 5.0 and on how digital transition can foster green transition. Sixth, research could focus on exploring new business models that leverage the capabilities of Industry 5.0, as well as examining the challenges and opportunities associated with their adoption. By pursuing research in these and other areas, scholars and practitioners can help to drive the development and implementation of Industry 5.0 and its associated technologies.

**Author Contributions:** Conceptualization, A.B.Y.; Methodology, A.B.Y. and I.M.; Writing—original draft, A.B.Y. and I.M. All authors have read and agreed to the published version of the manuscript.

**Funding:** This research received no external funding.

**Institutional Review Board Statement:** Not applicable.

**Informed Consent Statement:** Not applicable.

**Data Availability Statement:** Not applicable.

**Conflicts of Interest:** The authors declare no conflict of interest.

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
