# Peer review of "Linking Digital Technologies to Sustainability through Industry 5.0: A bibliometric Analysis"

_sustainability, doi:10.3390/su15097465_

Round 1

Reviewer 1 Report

The issues raised in the paper refer are very interesting and up-to-date. The aim of the research, posed by the author, was to conduct a bibliometric analysis to provide a comprehensive understanding of the geographical focus, methodological choices, prominent themes, and future research directions in the field of Industry 5.0 concept.  The authors reviewed 300 publications on Industry 5.0 to identify their theoretical roots, and research trajectories. The review was based on the integration of a co-citation analysis, historical direct citation analysis, and co-occurrence analysis. The authors in the abstract declared proposing new directions for research. However, I am of the opinion that in the course of the work carried out it was not possible, because the analyzes were bibliometric, rather than allowed for evaluating the content of the publications. Moreover, in the whole article the authors did not indicate the areas of future research.

Literature review

The authors stated the research question as follows:  “What are the implications, future prospects, and research areas of Industry 5.0?”. Is this research gap a result of the literature review? Did the authors manage to identify any research gap during the literature review? Please provide a strong research gap. Also please provide the reference to the sentence: “(…) we need an overview to understand recent developments in Industry 5.0 (trends, strengths, and gaps) – Introduction, lines 70-71

The author should also discuss whether there are any similar reviews published recently and, if yes, is this current review still relevant and of interest to the scientific community.

Research methodology

The query to the Scopus database was as follows: “Industry 5.0” OR “artificial intelligence” OR “smart manufacturing” OR “big data” OR “internet of things” OR “human-machine coexistence” OR “ Smart Sustainability” AND “Industry 5.0”. Why were these keywords chosen? They probably do not result from the review of the literature. Please provide an explanation. I suggest removing the first phrase “Industry 5.0” from the whole query as it is repeated at the end and it means that one of the queries checked incorrectly in the database was: “Industry 5.0 AND Industry 5.0”.

 The list of sources cited is not numerous and sufficient. Although most of them are quite up to date, but the authors should add some actual references to the bibliography.

Author Response

1

The authors in the abstract declared proposing new directions for research. However, I am of the opinion that in the course of the work carried out it was not possible, because the analyzes were bibliometric, rather than allowed for evaluating the content of the publications. Moreover, in the whole article the authors did not indicate the areas of future research.

We agree with the reviewer who we thank for the quality of his recommendations. In this bibliometric research, it was not possible to propose future research on industry 5.0 research, as the analyses were rather bibliometric and did not evaluate the content of publications. However, we have added a paragraph at the end of the text present some examples of the different research directions that could be pursued in Industry 5.0. By investigating these issues, researchers can help advance our understanding of this important area of industrial innovation. we define some axes of future research such as:

●      Human-machine collaboration

●      Twin transition

●      Sustainable production

●      Ethical considerations

2

The authors stated the research question as follows: “What are the implications, future prospects, and research areas of Industry 5.0?”. Is this research gap a result of the literature review? Did the authors manage to identify any research gap during the literature review?

We agree with the reviewer who we thank for the quality of his recommendations. In this bibliometric research, it was not possible to propose future research on industry 5.0 research, as the analyses were rather bibliometric and did not evaluate the content of publications. However, we have added a paragraph at the end of the text presenting some examples of the different research directions that could be pursued in Industry 5.0. By investigating these issues, researchers can help advance our understanding of this important area of industrial innovation. we define some axes of future research such as:

●      Human-machine collaboration

●      Twin transition

●      Sustainable production

●      Ethical considerations

3

The author should also discuss whether there are any similar reviews published recently and, if yes, is this current review still relevant and of interest to the scientific community.

We have added some recent studies.

4

“(…) we need an overview to understand recent developments in Industry 5.0 (trends, strengths, and gaps) – Introduction, lines 70-71

We have added a paragraph on this

5

The query to the Scopus database was as follows: “Industry 5.0” OR “artificial intelligence” OR “smart manufacturing” OR “big data” OR “internet of things” OR “human-machine coexistence” OR “Smart Sustainability” AND “Industry 5.0”. Why were these keywords chosen? Please provide an explanation.

We selected a diverse set of keywords for our bibliometric study, with the aim of expanding the research scope and including as many articles as possible related to Industry 5.0 as well as digital technologies that have contributed to recent technological advances and their impact on new industrial innovations.

6

The list of sources cited is not numerous and sufficient. Although most of them are quite up to date, but the authors should add some actual references to the bibliography.

We have added resources and reviewed existing resources in the text that were missing from the bibliography. As an example:

●    Javaid, M.; Haleem, A. (2020). Critical components of industry 5.0 towards a successful adoption in the field of manufacturing. Journal of Industrial Integration and Management. 5, 327–348.

●    Lu, Y.; Zheng, H.; Chand, S.; Xia, W.; Liu, Z.; Xu, X.; Wang, L.; Qin, Z.; Bao, J. (2022). Outlook on human-centric manufacturing towards Industry 5.0. Journal of Manufacturing Systems. 62, 612–627.

●    Madsen Øivind, D.; Berg, T. (2021). An Exploratory Bibliometric Analysis of the Birth and Emergence of Industry 5. Applied System Innovation. 4, 87. 

●    Aria, M., & Cuccurullo, C. (2017). bibliometrix: An R-tool for comprehensive science mapping analysis. Journal of Informetrics, 11(4), 959–975.

●    Bresnahan, T., Trajtenberg, M., 1995. General purpose technologies: ‘Engines of Growth’? Journal of Econometrics 65 (1), 83–108.

●    Cobo, M. J., López-Herrera, A. G., Herrera-Viedma, E., & Herrera, F. (2011). Science mapping software tools: Review, analysis, and cooperative study among tools. Journal of the American Society for Information Science and Technology, 62(7), 1382–1402.

●    Noyons, E.C.M., Moed, H.F., & Luwel, M. (1999a). Combining mapping and citation analysis for evaluative bibliometric purposes: A bibliometric study. Journal of the American Society for Information Science, 50(2), 115–131.

Reviewer 2 Report

Very interesting paper.  The paper points to journals that publish Industry 5.0 and the authors who deal with it. It would be interesting to analyze other databases (WOS) in this way and compare the results.

Figure 2 does not fully correspond with the text on lines 203-208, as the figure shows years from 2016 and the text mentions 2010.

Line 265: why the text mentions 2010 when the study covers the period 2016-2022 (Line 184).

Line 290: Figure 9, 14 - poorly readable.

Line 362: In the sentence you write that you point to four main findings, but you describe 5.

Figures 4-10  - it is not clear what each color represents

Author Response

to analyze other databases (WOS) in this way and compare the results.

We appreciate your recommendation. We have chosen Scopus as the only database: for the following reasons

Scopus has a wide coverage of research areas. Indeed, Scopus includes scientific journals from all disciplines, including the social sciences and humanities, while WOS focuses mainly on the natural and life sciences.

Second, Scopus is updated more frequently than WOS, which means that search results are more current.

Also, Scopus is updated more frequently than WOS, which means that the search results are more current. Finally, Scopus provides more complete data on authors, affiliations, and sources of information than WOS.

Figure 2 does not fully correspond with the text on lines 203-208 as the figure shows years from 2016 and the text mentions 2010.

Thank you so much for your comment. This is a typing error that we have corrected. It refers to 2020 and not 2010.

Line 265: why the text mentions 2010 when the study covers the period 2016-2022 (Line 184).

Thank you so much for your comment. This is a typing error that we have corrected. It refers to 2020 and not 2010.

Line 290 : Figure 9, 14 - poorly readable.

 We appreciate your feedback. As a result, we have improved the quality of a number of figures and eliminated five figures that added little value to the text. (Figure 4. Cross-country collaborations in industry 5.0 research. Figure 6. Network of contributing authors. Figure 7. Collaboration network among institutions.

Figure 9: co-citations to authors writing on industry 5.0 (using VOS viewer) and Figure 14: three field plot).

Line 362: In the sentence you write that you point to four main findings, but you describe 5.

Thank you for your comments. These are of course five fidings. We have corrected the sentence in the text.

Figures 4-10 - it is not clear what each color represents

The colors used in the figures indicate that authors, institutions, or countries sharing the same color may belong to a common cluster or have connections with each other in terms of publishing articles.

Reviewer 3 Report

I only have the following comments on the submitted article:

Using a large corpus of data (300 publications) from Scopus, this study conducts a bibliometric analysis to provide a comprehensive understanding of the geographical focus, methodological choices, prominent themes, and future research directions in this field.

I consider the submitted article to be up to date.

The current scientific literature is used in the article.

More literary resources could be used in the present article.

In the presented article, the images have poorer graphic quality and readability of values and it would be appropriate to improve them.

It is clear from the presented contribution that it will be necessary to continue the research.

The article would be appropriate to explicitly indicate the scientific benefits of the present article explicitly.

Author Response

1

More literary resources could be used in the present article.

We have added resources and reviewed existing resources in the text that were missing from the bibliography. As an example:

●    Javaid, M.; Haleem, A. (2020). Critical components of industry 5.0 towards a successful adoption in the field of manufacturing. Journal of Industrial Integration and Management. 5, 327–348.

●    Lu, Y.; Zheng, H.; Chand, S.; Xia, W.; Liu, Z.; Xu, X.; Wang, L.; Qin, Z.; Bao, J. (2022). Outlook on human-centric manufacturing towards Industry 5.0. Journal of Manufacturing Systems. 62, 612–627.

●    Madsen Øivind, D.; Berg, T. (2021). An Exploratory Bibliometric Analysis of the Birth and Emergence of Industry 5. Applied System Innovation. 4, 87. 

●    Aria, M., & Cuccurullo, C. (2017). bibliometrix: An R-tool for comprehensive science mapping analysis. Journal of Informetrics, 11(4), 959–975.

●    Bresnahan, T., Trajtenberg, M., 1995. General purpose technologies: ‘Engines of Growth’? Journal of Econometrics 65 (1), 83–108.

●    Cobo, M. J., López-Herrera, A. G., Herrera-Viedma, E., & Herrera, F. (2011). Science mapping software tools: Review, analysis, and cooperative study among tools. Journal of the American Society for Information Science and Technology, 62(7), 1382–1402.

●    Noyons, E.C.M., Moed, H.F., & Luwel, M. (1999a). Combining mapping and citation analysis for evaluative bibliometric purposes: A bibliometric study. Journal of the American Society for Information Science, 50(2), 115–131.

2

the images have poorer graphic quality and readability of values and it would be appropriate to improve them.

We appreciate your feedback. As a result, we have improved the quality of a number of figures and eliminated five figures that added little value to the text. (Figure 4. Cross-country collaborations in industry 5.0 research. Figure 6. Network of contributing authors. Figure 7. Collaboration network among institutions.

Figure 9: co-citations to authors writing on industry 5.0 (using VOS viewer) and Figure 14: three field plot).

3

It is clear from the presented contribution that it will be necessary to continue the research.

We agree with the reviewer who we thank for the quality of his recommendations. in this bibliometric research, it was not possible to propose future research on industry 5.0 research, as the analyses were rather bibliometric and did not evaluate the content of publications. However, we have added a paragraph at the end of the text present some examples of the different research directions that could be pursued in Industry 5.0. By investigating these issues, researchers can help advance our understanding of this important area of industrial innovation. we define some axes of future research such as:

●      Human-machine collaboration

●      Digital twins

●      Sustainable production

●      Ethical considerations

4

explicitly indicate the scientific benefits of the present article explicitly.

Many thanks for your comment, we have added the benefits of the paper!